# Electron tunneling at the molecularly thin 2D perovskite and graphene van der Waals interface

Kai Leng [1,2,8], Lin Wang [1,2,8], Yan Shao[1], Ibrahim Abdelwahab [1,2,3], Gustavo Grinblat[4], Ivan Verzhbitskiy [2,5], Runlai Li[1], Yongqing Cai[6], Xiao Chi[1,7], Wei Fu [1,2], Peng Song [1,2], Andrivo Rusydi[5,7], Goki Eda [1,2,5], Stefan A. Maier [3,4] & Kian Ping Loh [1,2✉]

Quasi-two-dimensional perovskites have emerged as a new material platform for optoelectronics on account of its intrinsic stability. A major bottleneck to device performance is the high charge injection barrier caused by organic molecular layers on its basal plane, thus the best performing device currently relies on edge contact. Herein, by leveraging on van der Waals coupling and energy level matching between two-dimensional Ruddlesden-Popper perovskite and graphene, we show that the plane-contacted perovskite and graphene interface presents a lower barrier than gold for charge injection. Electron tunneling across the interface occurs via a gate-tunable, direct tunneling-to-field emission mechanism with increasing bias, and photoinduced charge transfer occurs at femtosecond timescale (~50 fs). Field effect transistors fabricated on molecularly thin Ruddlesden-Popper perovskite using graphene contact exhibit electron mobilities ranging from 0.1 to 0.018 cm$^2$V$^{-1}$s$^{-1}$ between 1.7 to 200 K. Scanning tunneling spectroscopy studies reveal layer-dependent tunneling barrier and domain size on few-layered Ruddlesden-Popper perovskite.

[1] Department of Chemistry, National University of Singapore, Singapore, Singapore. [2] Center for Advanced 2D Materials and Graphene Research Centre, Singapore, Singapore. [3] Department of Physics, Imperial College London, London SW7 2AZ, UK. [4] Nanoinstitute Munich, Faculty of Physics, Ludwig-Maximilians-Universität München, 80539 München, Germany. [5] Department of Physics, National University of Singapore, Singapore, Singapore. [6] Institute of Applied Physics and Materials Engineering, University of Macau, Macau, China. [7] Singapore Synchrotron Light Source, National University of Singapore, 5 Research Link, 117603 Singapore, Singapore. [8] These authors contributed equally: Kai Leng, Lin Wang. ✉email: chmlohkp@nus.edu.sg

Two-dimensional (2D) Ruddlesden-Popper perovskites (RPPs) are natural quantum wells built of alternating layers of organic cations and inorganic anion cages. The appeal of RPP stems from its tunable chemical structure and dimensionality, which underpins its increasingly competitive performance in photovoltaics, light-emitting devices and ferroelectrics[1–6]. Heterostructure of 2D perovskites with other 2D materials is useful for fabricating functional devices[7,8]. Despite its promises in wide-ranging applications, making electrical devices on RPP is hampered by the high charge injection barrier imposed by the vertical array of organic cations[9,10]. For RPP crystals of the stoichiometric formulae $(C_4H_9NH_3)_2(CH_3NH_3)_{n-1}Pb_nI_{3n+1}$, the presence of multiple layers of dielectrics due to $(C_4H_9NH_3)^+$ causes poor interlayer charge transport, thus field-effect transistor (FET) devices are rarely reported. To date, FET mobilities of perovskites had been mainly reported for spin-coated 3D perovskites[11–14] and Sn(II)-based 2D perovskite films[15–17].

Recent work shows that layered RPP crystals can be exfoliated to yield flakes that are as thin as a single quantum well on account of the weak intermolecular van der Waals interactions between the organic chains[18]. The multiple quantum well structure of RPP presents thickness-controllable tunneling barriers, and the strong spin-orbit coupling (SOC) imbued in Pb, combined with a symmetry-breaking field in the direction perpendicular to the 2D surface, may generate Rashba spin-split bands, thus presenting the prospect for applications in spin-optoelectronics[19,20]. Compared to bulk crystals, molecularly thin perovskites offer better gate dielectrics and conformal adhesion on electrodes, thus mitigating the contact barrier issues faced by thicker crystals[18,21]. Identifying an electrode material with the interface energy alignment to favor low resistance contact on 2D perovskites is needed for the fabrication of FET[22]. Graphene has been used as an ohmic contact to semiconductor materials and also as a contact layer with gate-tunable barrier[23,24]. Due to its atomic

smoothness, graphene provides an atomically sharp interface that enables van der Waals assembly without the constraints of atomic commensurability[25–27]. However, little is known about this hybrid interface of 2D perovskite and graphene including their band alignment, the surface structure of 2D perovskites on graphene, and the dynamics of interfacial charge transfer.

Here, we investigate charge injection and electrical transport in molecularly thin single crystalline 2D $(C_4H_9NH_3)_2(CH_3NH_3)_3Pb_4I_{13}$ $((BA)_2(MA)_3Pb_4I_{13},$ $n=4)$ RPP using van der Waals-contacted graphene (G) as the electrode in FET devices. Control devices were fabricated on gold (Au) electrode. The atomic structure of the 2D perovskites on G and Au was examined using scanning tunneling microscope (STM), where a higher degree of substrate-induced disorder was found on Au than on G. Domain structure on molecularly thin 2D perovskites was observed to be thickness-dependent, and layer-dependent tunneling gaps were investigated using scanning tunneling spectroscopy (STS). A voltage-dependent transition from direct tunneling to Fowler-Nordheim (F-N) tunneling was observed at the RPP/G interface, in which the transition voltage can be gate-modulated. In addition, ultrafast charge transfer on the order of 50 fs or less between graphene and molecularly thin 2D perovskite was revealed by femtosecond pump-probe spectroscopy. Our studies demonstrate that molecularly thin 2D perovskite and graphene heterostructure may serve as an enabling platform for optoelectronics.

## Results

**Spectroscopic analysis of band edges and Fermi levels.** Unless otherwise stated, all the perovskites samples used in this study were $(BA)_2(MA)_3Pb_4I_{13}$ $(n=4)$ RPP (abbreviated as RPP hereafter). Figure 1a shows the schematic of the unit cell structure of RPP with four layers of inorganic quantum wells sandwiched by two layers of BA organic chains. Establishing the energy

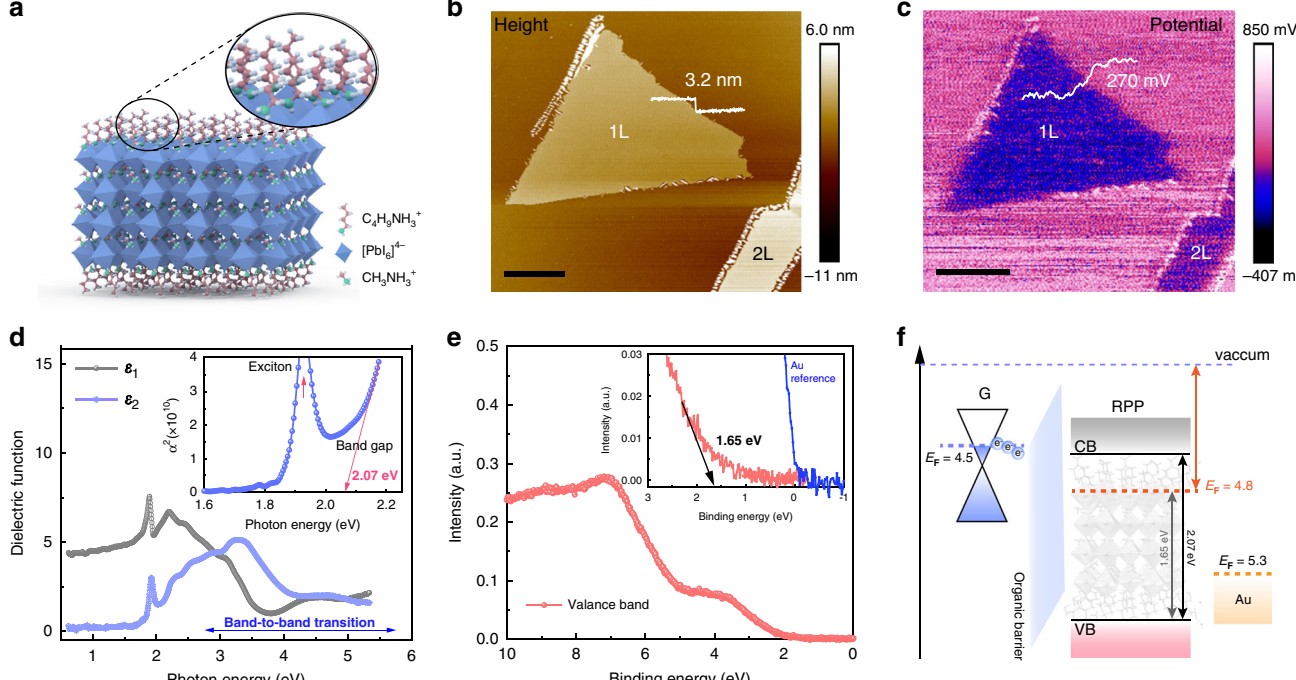

**Fig. 1 Spectroscopic analysis of Fermi level and band edges in Ruddlesden-Popper perovskite (RPP) relative to graphene (G) and gold (Au). a** Structural illustration of monolayer $n=4$ RPP $((C_4H_9NH_3)_2(CH_3NH_3)_3Pb_4I_{13})$. **b** Atomic force microscopy (AFM) image of monolayer and bilayer $n=.$ 4 RPP flakes exfoliated on pure silicon substrate. Scale bar, 1 μm. **c** Corresponding Kelvin probe force microscopy (KPFM) on the same area of (**b**). Scale bar, 1 μm. **d** Real ($\varepsilon_1$) and imaginary ($\varepsilon_2$) components of the dielectric function for $n=4$ RPP; inset shows the extrapolated electronic gap. **e** Valence band spectroscopy of $n=4$ RPP showing valence band edge. **f** Interfacial energy alignment diagrams of $n=4$ RPP, G and Au.

alignment between RPP and the contact electrodes (Au or G) provides the basic framework for understanding contact barriers and charge transfer behavior across the interface. For monolayer (1 L) and bilayer (2 L) $n = 4$ RPP flakes, the thickness has been determined by the atomic force microscope (AFM) to be 3.2 and 6.4 nm, respectively (Fig. 1b). The relative surface potential between the RPP flake and substrate was determined using scanning Kelvin probe microscope (KPFM) (Fig. 1c), and a work function ($\Phi$) of ~4.8 eV (Methods section) was calculated for both monolayer and bilayer RPP, thus $\Phi$ is thickness-independent. Furthermore, by performing KPFM across the RPP/G junction, the $\Phi$ of RPP was determined to be ~300 meV higher than the $\Phi$ of graphene (Supplementary Fig. 1). Combining the work function data with values of electronic band gap and band edges extracted from ellipsometry (Fig. 1d) and valence band spectroscopy measurement (Fig. 1e), an energy level alignment diagram of 2D RPP with respect to G and Au is illustrated in Fig. 1f. According to Fig. 1f, $n = 4$ RPP is $n$-type and its conduction band is near the Fermi level of G, thus electron injection from G to RPP should have a low barrier. On the contrary, the Fermi level of Au lies in the band gap of the $n = 4$ RPP, therefore, a larger Schottky barrier is expected at the RPP/Au interface than for RPP/G interface.

## Thickness-dependent surface domain change in RPP.

STM was used to study the atomic structure of the exfoliated perovskite flake on G and Au. It was observed that domains formed by paired structures on molecularly thin 2D RPP are configurable depending on the thickness of the RPP flakes. Figure 2a–c shows the STM empty state images of bilayer, monolayer, and thick RPP flakes on G. At positive substrate bias (+2.3 V, 10 pA), the image originates from electron tunneling into the conduction bands of Pb 6p orbitals, with additional contributions from the Pb 6s and I 5s according to DFT simulations (Supplementary Fig. 2) of the local density-of-states (LDOS). In the corner-sharing PbI$_6$ octahedral cage, the Pb atoms are at a lower position with respect to the iodine atoms, thus the latter is mainly imaged in STM[28,29]. At negative substrate bias (−2.3 V, 10 pA), the image is a convolution of electron tunneling from I 5p orbitals and organic molecules, and appears to be dynamically disordered owing to the coupling of tunneling electrons with the vibronic modes in the organic molecular layers, but when the bias is switched back to +2.3 V, the image reverts back to the initial ordered domain structure (Supplementary Fig. 3). For the sake of the present discussion, we focus our attention on the terminal iodine atoms of the octahedral cage. As measured from the STM image, the two nearest terminal iodine atoms at the surface are separated by 4.55 Å (Fig. 2a inset); this corresponds to a ~23.1% contraction of the unrelaxed unit cell length of 5.92 Å (Supplementary Fig. 4), indicating significant tilting of the terminal Pb-I bonds. These nearest neighbor iodine atoms formed a "paired" structure that is aligned unidirectionally with other pairs (Fig. 2a), such that collectively they form a single domain that can be distinguished from other domains that are 90 degrees rotated or mirror flipped (antiparallel) to it. Figure 2b shows multi-domain structure consisting of parallel and antiparallel domains for monolayer RPP on G. Changes in electrostatic boundary conditions, arising from polar discontinuity at surfaces and charge screening from the substrate, result in thickness-dependent change in the size and

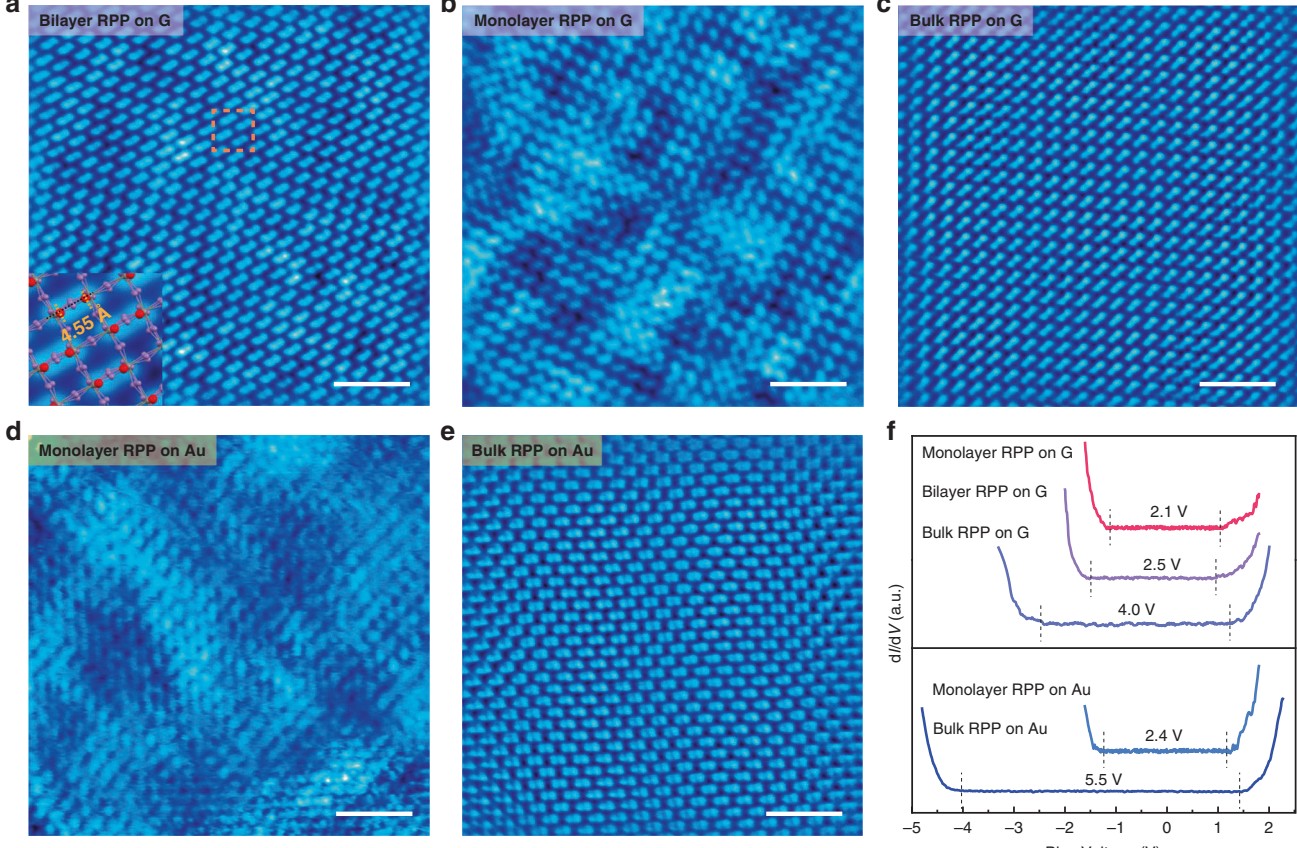

**Fig. 2 Scanning tunneling microscopy (STM) and spectroscopy (STS) on exfoliated $n = 4$ RPP flakes of different thicknesses on G and Au substrates.** Empty State (+2.3 V, 10 pA) images of **a** bilayer RPP on G; **b** monolayer RPP on G; **c** thick RPP on G. **d** Monolayer RPP on Au. **e** Thick RPP on Au. Scale bars, 4 nm (**a**–**e**). **f** STS (d$I$/d$V$) of $n = 4$ RPP flakes of different thicknesses on G and Au substrates.

orientations of the domains. STM studies reveal that the lateral width of the single domain increases with the thickness of RPP supported on G or Au (see below and Supplementary Fig. 5). For example, the domain width increases from ~15 nm in monolayer (Supplementary Fig. 5a) to ~100 nm in bilayer on G (Supplementary Fig. 5c). A completely disordered surface is seen for monolayer RPP on Au (Fig. 2d), in contrast to the single-domain structure seen for thick RPP on Au (Fig. 2e). The thickness-dependent surface domain ordering may originate from the dipolar screening of the internal electrostatic field in RPP by the substrate, or from the presence of substrate-induced strain, as both effects become increasingly stronger closer to the substrate[30,31]. Importantly, a higher degree of surface crystalline order was observed for RPP on G compared to Au, which is attributed to the high-quality van der Waals interface formed between RPP and G.

**Thickness-dependent tunneling gap change in RPP**. Next, the electronic structure of RPP flake from monolayer to bulk thickness is investigated. We measured the width of the energy gap in RPP layers of different thicknesses on G and Au (111) substrates by STS, in which the $dI/dV$ spectrum is proportional to the LDOS. The onset voltage at positive substrate bias is only slightly affected by the thickness of the RPP, whereas the onset voltage at negative bias shows a pronounced thickness-dependent shift (Fig. 2f). According to the LDOS of $n = 4$ RPP, the energy states due to the organic cations are 2.5 eV higher than the Pb $6p$ states at the conduction band. This implies that the organic chains are "transparent" during electron tunneling into the Pb-dominated conduction band states. Electron tunneling from the valence bands may leave behind holes, if these holes did not travel to the bottom electrode fast enough or get trapped, a space charge layer that is thickness-dependent may form, leading to the shift in the onset voltage. The apparent energy gap measured here is a convolution of the single electron band gap and tunneling barrier between the substrate and RPP. As shown in Fig. 2f, monolayer RPP on G has a smaller tunneling gap (~2.1 eV) than thicker flakes. For RPP flake of similar thickness, the energy gap is measured to be larger on Au than on G. Taking into account the higher quality domain order and smaller tunneling gap for perovskites supported on G than on Au, perovskite flakes with few-layer thickness (<10 nm), supported on G, were selected for device fabrication.

**FET devices based on molecularly thin 2D RPP**. To investigate the influence of contacts (G or Au) on the transport properties of 2D perovskite, we constructed bottom-contacted and bottom-gated FET devices with $n = 4$ RPP as the channel material (channel length = 300 nm). The G electrodes were pre-patterned on a $SiO_2$/Si substrate, followed by the dry transfer of a molecularly thin RPP flake that is two-unit cell thick onto the electrodes. The flakes were encapsulated with hexagonal boron nitride (h-BN) for protection against ambient (Fig. 3a inset). The use of single-crystalline molecularly thin perovskite flake allows a boundary-free domain to span across the channel. The FET devices were heated at 60 degrees for 5 min in a high vacuum before measurement. Device-to-device measurement shows good repeatability (Supplementary Fig. 6 and Fig. 7). Figure 3a shows the transfer characteristics of 2D RPP/G and 2D RPP/Au devices under scanning gate voltage ($V_g$) from −65 to 65 V at a fixed source-drain voltage ($V_{ds}$) of 3.5 V at 1.7 K. When no gate bias is applied, the devices are in the OFF state owing to the large thermal barriers for both electron and hole injections. For RPP/G FET, a much higher current is obtained when a positive gate bias is applied compared to negative gate bias, which is consistent with

the accumulation mode switching of the $n$-type device, attaining an ON/OFF ratio of $1.6 \times 10^6$ (Methods section). In contrast, the Au-contacted RPP device shows a source-drain current ($I_{ds}$) that is more than one order of magnitude lower compared with the G-contacted device. The electrode-dependent transport behavior can be explained by the position of the Fermi level of RPP relative to that of G (or Au), as illustrated in Fig. 1f. As $V_g$ increases in the positive direction in Fig. 3b, G becomes n-doped[32,33], which facilitates electrons transfer from G to RPP.

Besides the extrinsic factor of surface roughness on Au substrate, we believe the lower current on Au is due to the higher injection barrier on Au compared to G. By comparing output characteristics at 45–100 K for both G and Au-contacted RPP devices (Supplementary Fig. 8), the larger hysteresis present in the $I(V)$ curves of RPP/Au device may originate from the inhomogeneity of RPP on Au surface. The van der Waals interface between G and RPP allows conformal attachment and is beneficial for charge injection. Transfer curve recorded as a function of temperature exhibit a relatively constant hysteresis between 1.7–150 K in G-contacted FET (Fig. 3b). The hysteresis, which persists even at low temperature where Boltzmann's probability rules that ion or point defect migration is unlikely, may originate from a trap-dominated electronic mechanism[11]. In general, $I_{ds}$ decreases as the temperature is increased and field effect vanished above 250 K owing to screening of the gate field by ion migration, leading to very low mobility at $T > 200$ K (Supplementary Fig. 9). For an applied $V_{ds}$ in the range of 2 to 4 V and a $V_g$ in the range of −80 to 80 V (Supplementary Fig. 10), our best performing device shows a FET mobility of 0.1 $cm^2 V^{-1} s^{-1}$ recorded at 1.7 K on the reverse sweep, and 0.038 $cm^2 V^{-1} s^{-1}$ and 0.018 $cm^2 V^{-1} s^{-1}$ at 100 and 200 K, respectively (Methods section). These values are comparable with solution-processed 3D $MAPbI_3$ perovskites[34] (Supplementary Table 1).

The RPP/G interface is separated by both a van der Waals gap and a dielectric gap due to the presence of the molecular layers. To elucidate the charge injection mechanism across the interface, a F-N plot of $\ln(I/V^2)$ versus $1/V$ for the RPP/G junction at different $V_g$ and at 45 K is collected (Fig. 3c). The curve is extracted from the $I(V)$ output curve shown in Fig. 3c inset. A distinct change in transport mechanism from direct tunneling at a low bias to field emission at high bias[35] is observed for RPP/G at the transition voltage ($V_{trans}$); this clear transition is a hallmark of good interfacial contact. The value of $V_{trans}$ is related to the barrier height for charge injection. At a $V_g$ of 65 V, $V_{trans}$ is 0.50 V for the RPP/G interface compared with 1.53 V for RPP/Au interface at 45 K (Supplementary Fig. 11), thus evidencing a much lower barrier height on the RPP/G compared with RPP/Au device. Moreover, $V_{trans}$ is related to the energy offset between the electrode $E_F$ and the nearest molecular orbital (HOMO or LUMO)[36]. A decrease in the $| E_F-E_{LUMO} |$ offset can be explained by the elevated $E_F$ of G at a more positive $V_g$, resulting in a lowered tunneling barrier height. The linearly fitted slope at the F-N tunneling regime decreases as $V_g$ increases (Supplementary Table 2), which agrees with the trend of $V_{trans}$. Figure 3d shows a 2D color map of $d\ln(I/V^2)/d(1/V)$ as a function of $V$ and $T$ under a $V_g$ of 65 V. The white line defines the transition voltage between direct (left) and F-N tunneling (right). $V_{trans}$ shifts to higher bias as temperature increases, this is accompanied by a decreasing $I_{ds}$ (Supplementary Fig. 12), which is attributed to the screening of the gate electrostatic field by ion migration.

**Photo-FET study based on molecularly thin 2D RPP**. Upon photo-irradiation with energy larger than the quasi-particle gap (focused laser with 532 nm excitation; 1 $\mu m^2$ spot area), the output curve of RPP/G photo-FET device shows a non-linear

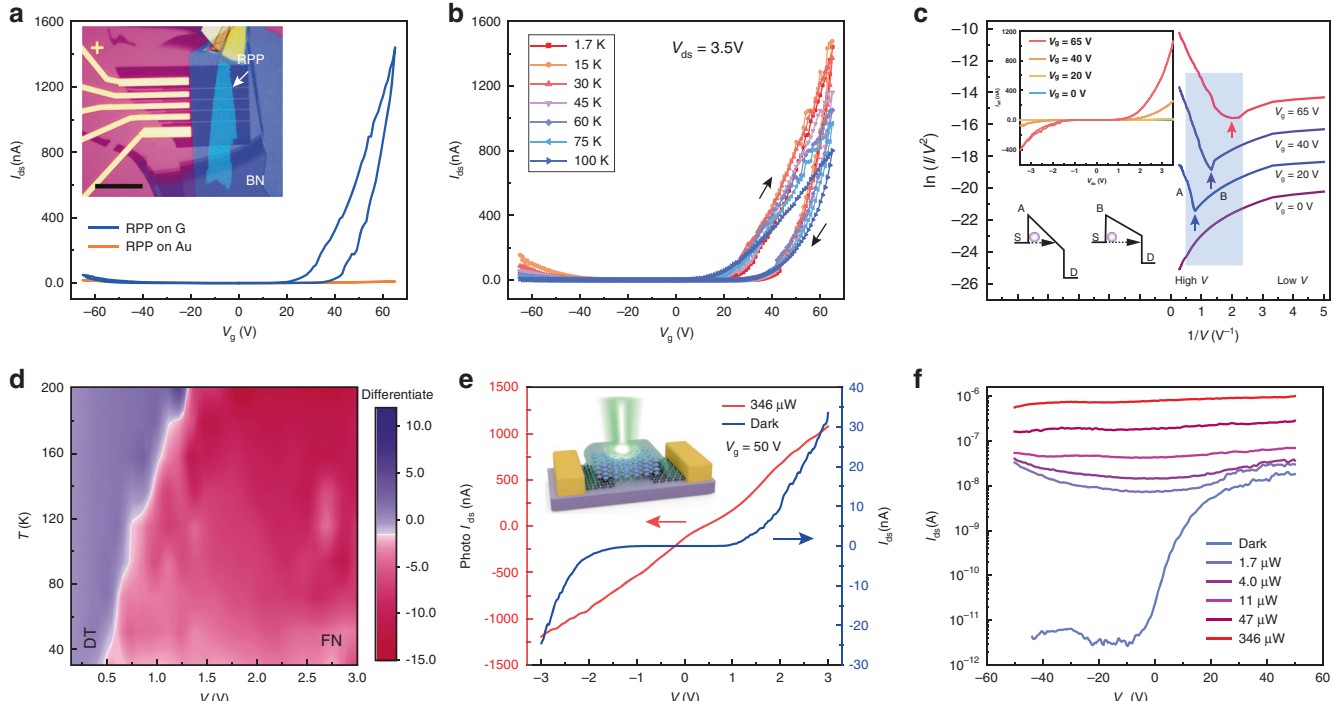

**2D perovskite FET devices ($n = 4$) and their photo-electrical characterizations, using G or Au as electrodes.** Fig. 3 **a** Transfer characteristics of RPP/G and RPP/Au FET devices at 1.7 K. Inset, Optical image of a typical bilayer thick 2D RPP device using G as electrode. Source-drain channel length is 300 nm. Scale bar, 15 µm. **b** Temperature-dependent transfer characteristics at 1.7–100 K for RPP/G FET. **c** Fowler-Nordheim (F-N) plots of 2D RPP/G, with each plot collected at a fixed gate voltage. The minimum voltage highlighted by the arrow is the transition point from direct tunneling (DT) to F-N as voltage increases. Schematic diagrams corresponding to F-N tunneling (point A) and DT (point B). Inset, $I(V)$ curves corresponding to the F-N plot. **d** Temperature-dependent color map of F-N plots. The white line indicates the transition boundary between DT and F-N tunneling. **e** Output characteristics in the dark and under laser illumination based on 2D RPP/G FET device at 77 K. **f** Transfer characteristics in the dark and under different illumination intensities for RPP/G device at 77 K.

(blue)-to-linear $I$-$V$ behavior (red) transition, as shown in Fig. 3e and Supplementary Fig. 13a. Photo-irradiation creates photocarriers and reduces the Schottky barrier width, thus reducing interfacial resistance[37]. A similar photoinduced transition also happened at $V_g = -50$ and 0 V (Supplementary Fig. 13b and c). Figure 3f displays the transfer characteristics of the phototransistor as a function of illumination intensities, where the carrier concentrations have increased a thousand-fold upon photoexcitation. Both electron or hole concentrations were enhanced significantly with increasing laser power density, and the conductance gap decreases drastically. The big increase in photocurrent over dark current is the main reason for the high photoresponsivity of photodetector fabricated using 2D RPP[18].

**Ultrafast charge transfer in heterostructure of RPP/G.** The dynamics of photocarrier injection across the RPP/G interface was studied by femtosecond pump-probe spectroscopy using sub-10 fs pulses. The experiment was performed by pumping the sample with a broad 600–750 nm wavelength beam at 25 pJ/µm² peak energy density, and using a 750–1000 nm wavelength beam as the probe (Methods section). The analysis reveals ultrafast injection of photo-excited carriers from G to RPP at timescales <100 fs, approaching the fastest limit of charge transfer reported for van der Waals-stacked interface[38]. The differential reflectivity ($\Delta R/R$) results measured from $n = 4$ RPP flake on glass, graphene on glass, and RPP/G heterostructure on glass, respectively, as a function of probe wavelength ($\lambda$) and pump-probe delay time ($t$), within the first 1000 fs after pump arrival, as shown below. A positive $\Delta R/R$ signal indicates enhanced reflectivity, while a negative signal indicates the opposite. It can be noticed that the

bare RPP flake shows a faint negative response close to $t = 0$ fs (Fig. 4a), presumably coming from nonlinear changes in its complex refractive index due to the optical Kerr effect and two-photon absorption[39]. In contrast, G (Fig. 4b) and RPP/G heterostructure (Fig. 4c) present a strong positive signal. Comparing the time response between G and RPP/G reveals a much faster decay in the $\Delta R/R$ signal in the heterostructure. This is clearly reflected in Fig. 4d, which exhibits the data for the two cases at $\lambda = 820$ nm. A two-term exponential decay function ($y = A\exp(t/\tau_1) + B\exp(t/\tau_2)$) was used to fit the data at $t > 0$ fs. The obtained characteristic decay times for graphene are $\tau_1 < 11$ fs (limited by the instrument response function, IRF), and $\tau_2 = (300 \pm 30)$ fs. $\tau_1$ is assigned to intraband carrier thermalization by carrier–carrier scattering, while $\tau_2$ is attributed to carrier–phonon scattering. Remarkably, in the case of the G-RPP heterostructure, the differential reflectivity signal vanishes within a 100-fs time period. For this case, we find that $\tau_1$ coincides with the value for bare G, while $\tau_2$ shortens to only ~50 fs, indicating ultrafast charge transfer from the G layer to the RPP flake. Besides the energy level matching between graphene's Fermi level with the conduction bands in RPP, the ultrafast charge transfer may be enabled by photo-assisted quantum tunneling across the RPP/G interface.

**Discussion**

In conclusion, our studies show that graphene serves as a low barrier and gate-tunable van der Waals contact for $n = 4$ RPP where we observed the classic tunneling mechanism of field emission at high bias voltage and direct tunneling at low bias. A considerably lower barrier for charge injection was observed on

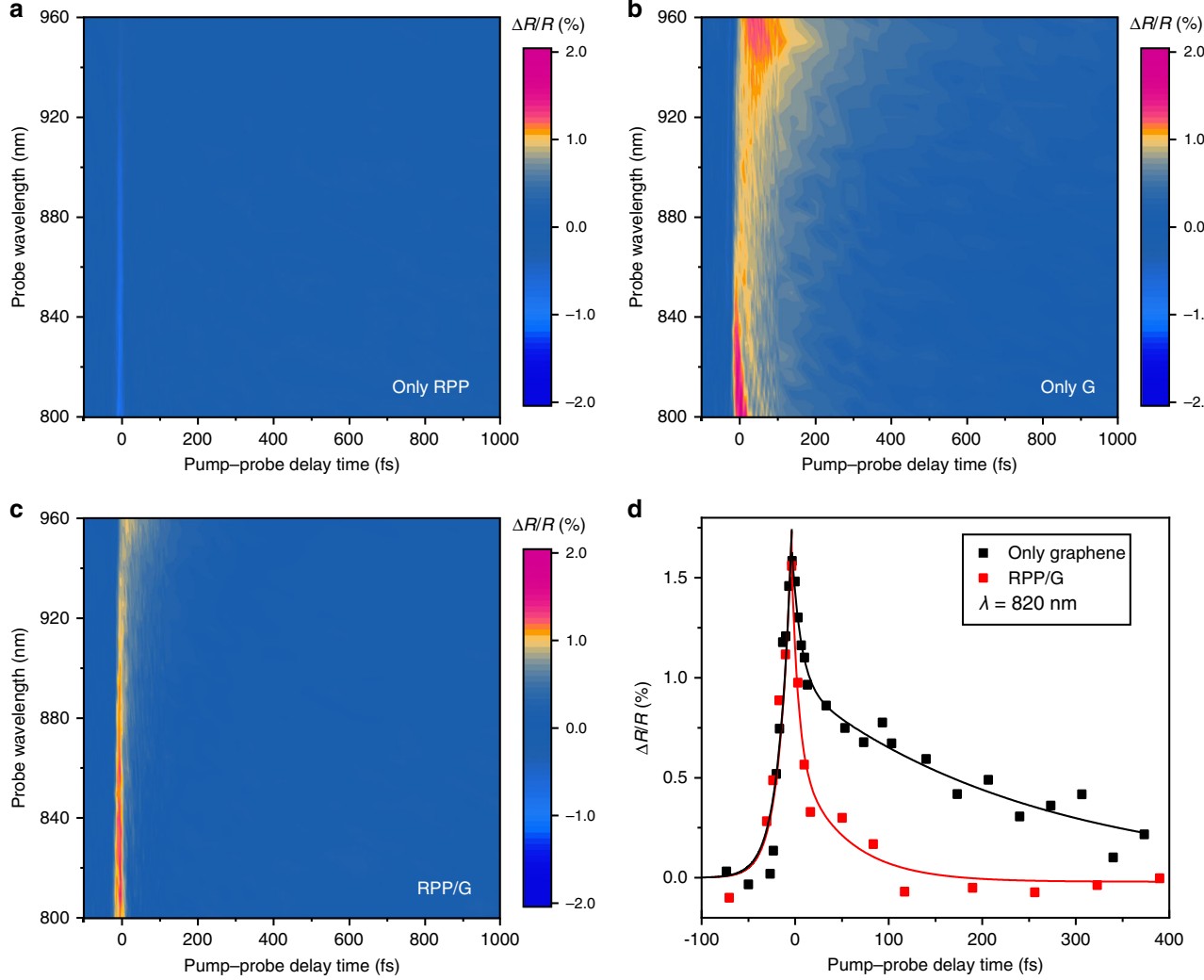

**Fig. 4 Femtosecond pump-probe study of dynamic photocarrier injection across RPP/G heterostructure. a–c** Differential reflectivity spectra of (a) $n = 4$ RPP flake, **b** G, and **c** RPP/G heterostructure as a function of pump-probe delay time and probe wavelength, when pumping at 25 pJ/μm² peak energy density. **d** Cross-section of the data plotted in (**b** and **c**) at $\lambda = 820$ nm. Solid lines in the graph correspond to fits considering exponential growth ($t < 0$ fs) and decay ($t > 0$ fs) functions.

RPP/G as compared to the RPP/Au interface. The quality of the interface was further verified by ultrafast charge transfer at the RPP/G interface (~50 fs), which occurs at a timescale similar to transition metal dichalcogenide heterostructures showing a type II energy offset. Highly ordered domains were observed under the STM for ultrathin RPP on graphene, where the thickness-dependent tunneling gap was observed. Our STM studies show that depending on the substrates, polar domains re-order differently on molecularly thin perovskites owing to changes in electrostatic boundary conditions. This has implication for Rashba type spin-splitting, which has been predicted to be enhanced for structures which are inversion asymmetric. The presence of an ordered array of organic chains interleaving perovskite crystals hints at an interesting, yet unexplored possibility of using molecularly thin RPP both as both a tunneling barrier and spin generator on graphene.

## Methods

**Synthesis of $(C_4H_9NH_3)_2(CH_3NH_3)_3Pb_4I_{13}$ ($BA_2MA_3Pb_4I_{13}$; $n = 4$) bulk single crystal**. A temperature-programmed crystallization method was applied to synthesize $n = 4$ RPP single crystal[18]. PbO (0.69 M), BAI (0.17 M), and MAI (0.52 M) precursors were dispersed in a concentrated HI and $H_3PO_2$ mixture (7.6:1, vol/vol) in an Ar-filled glove box, and then heated at 110 °C with stirring for 40 min to give

a clear yellow solution. The solution was quickly transferred to an oven at 110 °C and allowed to cool slowly to room temperature at a rate of 3 °C h⁻¹, whereupon metallic black square- or rectangle-shaped crystals started to form. The crystals were isolated by vacuum filtration and dried in an Ar-filled vacuum chamber at room temperature.

**Fabrication of 2D RPP FETs on exfoliated thin flake**. Because perovskite is sensitive to polar solvent, dry transfer was applied to fabricate perovskite-based device. To fabricate G or Au-contacted RPP devices, G or Au electrode was firstly patterned on a $SiO_2/Si$ substrate. The exfoliated RPP thin flake was then dry transferred onto these electrodes. Finally, h-BN encapsulation was a very necessary step to protect the RPP channel from degradation and maintain device stability during the test.

**Work function**. Two methods based on $n = 4$ RPP exfoliated on different substrates (pure Si and G) were carried out to calculate work function of $n = 4$ RPP. The results were consistent. Method one: Monolayer and bilayer $n = 4$ RPP flakes were first exfoliated onto pure Si substrate as shown in Fig. 1b. Then KPFM measurement (Bruker Multimode 8) was conducted on these flakes to get relative surface potential between RPP and Si (Fig. 1c). An external voltage was applied to the tip with the sample grounded. In this scenario, a negative KPFM signal indicates a higher work function of the sample relative to the probe. Using the pre-calibrated Φ of Si as 4.6 eV, the Φ of monolayer n = 4 RPP ($\Phi_{1L\ RPP}$) will be 4.6 + 0.27 = 4.87 eV. The calculated Φ of bilayer $n = 4$ RPP ($\Phi_{2L\ RPP}$) is almost same as monolayer. Method two: Monolayer, bilayer, and trilayer $n = 4$ RPP flakes were exfoliated onto G substrate (Supplementary Fig. 1). This allows us to evaluate the work function difference between $n = 4$ RPP and G by KPFM measurement

(Bruker Dimension Icon). Different from method one, here the external voltage was applied to the sample with the tip grounded. In this case, a positive KPFM signal indicates a higher work function of the sample relative to the probe. If the $\Phi$ of G is 4.5 eV as reported[40,41], $\Phi_{1L\ RPP} = 4.5 + 0.34 = 4.84$ eV. $\Phi_{2L\ RPP} = 4.5 + 0.31 = 4.81$ eV, $\Phi_{3L\ RPP} = 4.5 + 0.34 = 4.84$ eV. We are pleased to see that both methods yield $\Phi_{RPP} = 4.8$ eV.

**On/off FET current ratio**. The RPP/G FET exhibits a low drain current of $8.5 \times 10^{-13}$ A at $V_g$ of 5 V and a drain current of $1.4 \times 10^{-6}$ A under $V_g$ of 65 V. The on/off ratio is calculated to be $1.6 \times 10^6$.

**Mobility extraction**. In the work, as the applied $V_{ds}$ (<4 V) is much smaller than the $V_g$ sweep range (−80 to 80 V or −65 to 65 V), the equation of $\mu = L/WC_iV_{ds} \cdot dI_{sd}/dV_g$ was used to extract device mobility[42], where $L$, $W$, and $C_i$ are the length of the channel, the width of the channel, and the gate capacitance per unit area, respectively. For example, for the device shown in Fig. 3a of the main text, $L$ is 300 nm, $W$ is 6.5 μm, $dI_{ds}/dV_g = 1.03 \times 10^{-7}$ A/V at 1.7 K for a $V_{ds}$ of 4 V (Supplementary Fig. 10b). Given that $C_i = 11.5$ nF/cm², the calculated carrier mobility is 0.1 cm²/Vs. Based on the same calculation method, when the RPP/G device was tested under $V_g$ of 65 V and $V_{ds}$ of 3.5 V at 100 and 200 K (Supplementary Fig. 9), the calculated mobilities were 0.038 and 0.018 cm²/Vs, respectively.

**Injection barrier height estimation**. At high $V_{ds}$, the $J$-$V$ relation is described by F-N tunneling:

$$I \propto V^2 \exp\left(-\frac{4d\sqrt{2m_e\Phi^3}}{3\hbar qV}\right) \tag{1}$$

where $d$, $m_e$, $\Phi$, $\hbar$, and $q$ denote barrier width, effective electron mass, barrier height, reduced Planck's constant, and elementary charge, respectively. Therefore, a plot of $\ln(I/V^2)$ against $1/V$ gives $4d\sqrt{2m_e\Phi^3}/3\hbar q$ from the slope. In the work, the barrier width $d$ was assumed to be the length of BA organic chain (6.3 Å), $m_e$ is $9.1 \times 10^{-31}$ kg, and the $\ln(I/V^2)$ against $1/V$ is −1.86 A/V under $V_g$ of 65 V at 1.7 K (Supplementary Fig. 12c) in a graphene-contacted FET. Based on these, a charge injection barrier $\Phi$ of 0.63 eV at 1.7 K was obtained. By the same calculation method, the Au-contacted RPP device showed a $\Phi$ of 1.82 eV at 1.7 K.

**Spectroscopic ellipsometry measurement**. The ellipsometry studies were carried out on J. A. Woollam ellipsometer which has monochromatic radiation photon with energies ranging from 0.6 to 6.5 eV. The sample is measured in high vacuum cryostat with pressure of $1 \times 10^{-9}$ Torr with incidence angles ($\theta$) at 70°. Spectroscopic ellipsometry directly measures the changes of the polarization states, which can be expressed by two parameters $\psi$ and $\Delta$. Then the Fresnel reflection coefficient ratio ($\rho$) between the two polarized (s- and p-) ($r_p$ and $r_s$) light can be obtained by calculating the Eq. (2) as following[43,44]:

$$\rho = r_p/r_s = \tan\Psi e^{i\Delta} \tag{2}$$

where $\Delta$ is the change in phase and $\psi$ is the intensity ratio for p- and s- components of the light after it has interacted with the sample. For the bulk materials, the real and imaginary parts of the complex dielectric function $\varepsilon_1$ and $\varepsilon_2$ can be directly extracted from the Fresnel equations, which are defined as

$$r_p^{ab} = \frac{n_b\cos\theta_a - n_a\cos\theta_b}{n_b\cos\theta_a + n_a\cos\theta_b} \tag{3}$$

$$r_s^{ab} = \frac{n_a\cos\theta_a - n_b\cos\theta_b}{n_a\cos\theta_a + n_b\cos\theta_b} \tag{4}$$

$$\sqrt{\varepsilon(\omega)} = n(\omega) \tag{5}$$

where $n$ and $\theta$ represent the complex refractive index and incidence (transmission) angles, respectively. $a$ and $b$ signify the ambience and bulk sample for the bulk modeling.

**Photoemission spectroscopy measurement**. PES experiments were performed at SINS beamline of Singapore Synchrotron Light Source in an ultrahigh vacuum chamber with a base pressure $<1 \times 10^{-10}$ Torr. The energy resolution is better than 50 meV for valance band measurement. The binding energy of samples is calibrated and referenced to the Fermi level of a sputtered gold foil.

**AFM/KPFM measurement**. Bruker Multimode 8 AFM machine in peak force mode and Bruker Dimension Icon in peak force mode AFM machine were used to carry out morphology and KPFM measurement on $n = 4$ RPP flakes. Data in Fig. 1b and c are measured by Multimode 8 machine in glove box. Data in Supplementary Fig. 1 are measurement by the Dimension Icon machine in air.

**STM/S measurement**. STM and STS measurements were carried out under ultrahigh vacuum with a base pressure of $2 \times 10^{-11}$ mbar at 4.5 K using a commercial Omicron LT STM system. All STM images were collected under constant-current mode with electro-chemically etched tungsten tips and all given voltages refer to the sample. The differential conductance (d$I$/d$V$) spectra were collected by using standard lock-in techniques with a voltage modulation of 10 mV and frequency of 973 Hz. Before real-sample STS measurements, the tip was calibrated on a clean Au (111).

**Temperature-dependent electrical measurement**. The transfer and output characteristics were carried out in Oxford Teslatron system from temperature of 1.5–273 K with a vacuum of mbar. Transfer curves were recorded with Keithley 2401 sourcing a constant DC voltage between source-drain contacts and Keighley 6430 sourcing a sweeping gate voltage. Output curves were recorded with Keithley 2401 sourcing a sweeping voltage under constant gate voltage sourced by Keighley 6430.

**Femtosecond pump-probe measurement**. The ultrafast pump-probe experiments were performed using ~7 fs pulses at 100 kHz repetition rate. The pump and probe beams were composed by ~600–750 nm and ~750–1000 nm spectral components, respectively, and were focused onto the sample using a metal objective of 0.5 numerical aperture. The measurements were carried out with lock-in detection by modulating the pump beam using an optical chopper at <1 kHz frequency. The time difference between the pump and probe pulses was controlled with a motorized optical delay line with <1 fs accuracy. A spectrograph coupled to a low-noise Si photodiode was used for spectral characterization of the probe light reflected by the sample.

**First-principles computational methods**. Energetics and STM simulation on nonpair/pairing structures. Simulation of the pairing tendency was performed with building $\sqrt{2} \times \sqrt{2}$ pseudocubic supercells of the RPP. The pseudo-symmetry was broken by deviating the surface I atoms from respective bulk positions to form pairing structure with their direct neighbors, followed by fully structural relaxations. The STM image under a specific voltage ($V$) was simulated following the scheme proposed by Tersoff and Hamann[45] with $\rho_{STM}(\mathbf{r}, V) =$

$$\int_{E_f - eV}^{E_f} \sum_{n,\mathbf{k}} \psi_{n,\mathbf{k}}(\mathbf{r})\delta(E - E_{n,\mathbf{k}})dE$$

where $E_f$ is the Fermi level, $\psi_{n,\mathbf{k}}(\mathbf{r})$ are the eigenstates of the unperturbed surface with eigenvalues $E_{n,\mathbf{k}}$ both are derived by solving the Kohn-Sham eigenstates using density functional theory.

## Data availability

The data sets generated and/or analyzed during the current study are available from the corresponding author on reasonable request.

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

## Acknowledgements
K.P.L. thanks funding support from Singapore's Ministry of Education Tier 2 fund entitled "Rashba Effect and Ferroelectricity in Two-Dimensional Hybrid Perovskites", project number MOE2019-T2-1-037. G.E. acknowledges support from the Ministry of Education (MOE), Singapore, under AcRF Tier 3 (MOE2018-T3-1-005). S.A.M. acknowledges the Lee-Lucas Chair in Physics and the DFG Cluster of Excellence EXC 2089/1-390776260 e-conversion.

## Author contributions
K.L. and K.P.L. conceived and designed the work; K.L. fabricated RPP single crystals; L.W. and K.L. fabricated high quality G electrodes; K.L. exfoliated and made FET devices; K.L. tested FET devices; K.L. and W.L. did FET data analysis; Y.S. performed STM and STS measurement on samples prepared by K.L.; I.V. and K.L. tested photo-FET device guided by G.E.; G.G. and I.A. performed femtosecond pump-probe measurements guided by S A.M.; R.L.L. helped to design and draw feature image of this work; Y.Q.C. performed calculations; X.C. conducted ellipsometry and valence band spectroscopic measurements guided by A.R.; W.F. helped to improve the drawing; P.S. helped to load device; K.L., L.W., and K.P.L. wrote the manuscript; All authors contributed to the overall scientific interpretation.

## Competing interests
The authors declare no competing interests.
