## [Peer Review File · Nature Communications]

REVIEWER COMMENTS

Reviewer #1 (Remarks to the Author):

In this manuscript, the authors employed graphene as the electrical contact for RPP halide perovskites ($n=4$). Compared with metal contacts, graphene is found to possess a lower barrier for charge injection. Field effect transistors (FET) utilizing RPP/graphene contacts exhibit relatively good electron mobilities. This work can be published in Nat. Commun. if the following questions and comments can be addressed.

1. The authors mentioned that Fig. 2b shows a 180 degree-rotated antiparallel domain structure between the bright and dark contrast regions for monolayer. However, the definition of 180 degree is not very clear because the bright domain and dark domain have an angle of 90 degree.

2. The RPP perovskite electronic devices with graphene as the electrical contact had a much better performance than those with Au as the electrodes. In addition to some intrinsic differences, including the different work functions and tunneling gap between graphene and Au, the extrinsic factors, such as the different surface roughness and thickness between graphene and Au, may also contribute to the variation in device performance. Typically, Au has a rougher surface than atomically flat graphene, which will give rise to a poorer contact quality. And usually the thickness of Au electrodes is around 50-100 nm. Considering the fragile mechanical properties of RPP halide perovskite, it is very easy to generate some defects, sometimes even fractures, in RPP perovskites when transferring a few layers of RPP perovskites onto pre-patterned Au electrodes. Please elaborate the role from the extrinsic factors in the main text, and discuss which should be the dominating factor in determining the performance of the related RPP perovskite devices.

3. It seems that the deviation between devices is significant. As was claimed by the authors, there should be a clear Schottky barrier between graphene and RPP. However, the barrier seems to disappear in Extended Data Fig.11a. In addition, please explain why there is an open-circuit voltage and short circuit current for the red curve in Extended Data Fig.11b (346 μ W, $V_g=-50$ V).

4. As shown in Extended Fig 11c, the RPP/graphene devices is very insulating even under a very strong illumination of 346 μ W. This is much lower than the RPP/Au devices the authors' previous report in Nature Materials. Based on this comparison, the Au seems to have much better contact than graphene, which is opposite to the conclusion for this manuscript.

5. From the discussion in the main text and Fig. 1f, the $n=4$ RPP halide perovskite is highly n doped as the fermi level is very close to the conduction band minimum. However, this is counterintuitive as the intrinsic electrical conductivity of the $n=4$ RPP halide perovskites without gating and illumination is very low (from this manuscript and previous reports). If possible, please estimate the carrier concentration of $n=4$ RPP halide perovskite based on the electrical measurement results.

6. The reference citations are out of date. Several highly relevant works are not mentioned in the introduction. For example, regarding the graphene and 2D perovskite heterostructure, a recent electrical study (Joule 2018, 2, 2133) and a comprehensive review (Chemical Society Reviews 2018, 47, 6046-6072) should be cited. Regarding charge injection from Au electrode to 2D perovskites and related FET performance, a recent studying incorporating conjugated organic ligands (JACS 2019, 141, 15577-15585) should be mentioned.

7. Please double check the maximum laser power density, which was 346 μ W in the figures but 348 μ W in the main text.

Reviewer #2 (Remarks to the Author):

The manuscript reports the study of the 2D RPP/graphene heterojunction through FET performance, STM/STS and femtosecond pump-probe characterization. It contains some interesting results, with the STM studies showing distinct morphology on different substrates. However, it is not clear at this point what changes the FET device performance as the graphene and Au contact are prepared using different techniques, note the top surface of Au after deposition can be rather rough. So based on the current data, it is hard to say whether the differences come from the van der Waals contact or just because of the change in metal work function. The control experiment present in the work contains two varying factors. For a control experiment, it would be more proper to use Ag whose work function is close to 4.5eV. On the other hand, although the graphene contact shows lower tunneling barrier compared with Au, which is -0.6eV , but the value itself seems to have room for improvement.

Below are more specific questions:

- 1) How is the graphene Fermi level determined to be 4.5eV?
- 2) It is strange that the Fermi level doesn't change with layer number at all but the band gap varies with layer number as shown in the STS measurements. Based on the band analyses in Fig. 1, the monolayer RPP should have fairly high electron density so that narrow tunneling barrier width, what is the basic two-terminal IV looks like under dark.
- 3) Considering the large hysteresis in the FET transfer curve, how is the mobility determined under such circumstances. And what is the more fundamental reason for FET mobility increase with decreasing temperature.
- 4) The authors mentioned "The value of V_{trans} is related to the barrier height for charge injection", but as the slope of Fig. 3c doesn't change between each curve, so the barrier height seems to be constant, the authors need to explain this part.
- 5) Comparing Fig. 3b and 3f, the 75K dark transfer data showing at least an order difference, please explain the discrepancy. Why does sample under illumination show weak field effect, especially for weak illumination case where the current is not large (i.e. the carrier density is not high). Additionally, the illumination intensity is suggested to be normalized in the format of power/area, which would allow the readers to compare with others' work easier.
- 6) Extended Fig.6 suggests G/Au based devices show different asymmetries in the transfer characteristics. Does it mean the carrier type is different, please explain.
- 7) Based on the current analyses, it is hard to evaluate the performance of the 2D RPP/graphene FET of this work with other lead halide perovskite FETs reported in the literature, a table including on/off ratio, on state current, geometry, FET mobility.... is suggested to be added.
- 8) Please mark the domain sizes in Fig. 2 based on the discussion "on graphene or Au (Fig. 2c and 2d) can have a single domain with lateral width $>200\text{ nm}$, but the size of the domain decreases to 100 nm and 20 nm in bilayer and monolayer (Fig 2a and 2b)".

Reviewer #3 (Remarks to the Author):

In this manuscript, Leng and coauthors introduce graphene to contact molecularly thin 2D perovskite plane so that the contact barrier is reduced and the mobility of 2D perovskite is improved. Due to the presence of long organic chains in 2D perovskites, the electrical conductivity in the vertical direction is rather poor, leading to the poor performance for 2D perovskite based electronic devices. The lowered contact barrier in 2D perovskite/graphene interface thus can

partially solve the charge injection issue in 2D perovskite electronic devices. In particular, STM study reveals that highly ordered domain structure can be sustained down to a monolayer for 2D perovskite plate on graphene while disorders can be observed for monolayer 2D perovskite plate on Au. This study is interesting and important for 2D perovskite community. However, the following my concerns are needed to addressed before I can recommend its publication.

1 The authors claimed that from STM study, a thick 2D perovskite crystal supported on graphene or Au can have a single domain with lateral width >200 nm, but the size of the domain decreases to 100 nm and 20 nm in bilayer and monolayer. Is it possible that even in thick sample the size of the domain is still around 20 nm like monolayer case since the interlayer coupling in 2D perovskite is rather weak? Also, can the authors comment how the domain size affects the transport properties of the graphene contacted 2D perovskite devices?

2 In Figure 1d, the authors extracted the bandgap of 2D perovskite by extrapolated the edge of absorption spectrum. However, due to the strong excitonic effect, this method might be not applicable here since the exciton states with large n would also contribute to the absorption edge.

3 Can the authors comment how the degradation of 2D perovskite affects their results? Previous study has revealed that despite the stability of 2D perovskite has been improved, they still undergo quick degradation. Under such case, it is evitable that degradation would take place during the dry transfer process. With the degradation of 2D perovskite, surface depletion field might be present (see Nano Research 12, 2858–2865(2019)). The surface depletion field possibly effect the STM image.

4 The authors attribute the hysteresis in transfer curves to the trap-dominated electronic mechanism while no field effect above 250 K to the screening of ion migration. Under such case, how did the authors exclude the influence of ion migration on the hysteresis.

5 Since a large hysteresis has been observed, the authors are needed to specific how did they extract mobility. A different mobility will be obtained from backward and forward scanning. In addition, the authors are suggested to comment how the hysteresis would affect the precise evaluation of mobility.

6 Apparently, the contact is non-Ohmic either in dark or under light illumination.

7 How the light illumination affects the hysteresis in transfer curves?

8 The authors attribute the increase of current under light illumination to the exciton dissociation. Nevertheless, the exciton binding energy is estimated to be over 100 meV from Figure 1d. Under such case, the exciton dissociation cannot occur at room temperature.

9 Have the authors measured the device performance of graphene contacted monolayer 2D perovskite?

10 Can the authors comment what will change if we use different type of long organic chain in 2D perovskites?

Response to reviewer 1

Reviewer #1 (Remarks to the Author)

In this manuscript, the authors employed graphene as the electrical contact for RPP halide perovskites ($n=4$). Compared with metal contacts, graphene is found to possess a lower barrier for charge injection. Field effect transistors (FET) utilizing RPP/graphene contacts exhibit relatively good electron mobilities. This work can be published in Nat. Commun. if the following questions and comments can be addressed.

1. The authors mentioned that Fig. 2b shows a 180 degree-rotated antiparallel domain structure between the bright and dark contrast regions for monolayer. However, the definition of 180 degree is not very clear because the bright domain and dark domain have an angle of 90 degree.

ANS: We apologize for the mistake; instead of 180 degree, it should be 90 degree. Action: we have revised the relevant lines in the paper accordingly:

In page 5, “These nearest neighbour iodine atoms formed a “paired” structure that is aligned unidirectionally with other pairs, such that collectively they form a single domain that can be distinguished from other domains that are antiparallel (90 degree rotating or mirror flipping) to it.”

“Fig. 2b shows multi-domain structure consisting of parallel and antiparallel domains for monolayer RPP on G.”

2. The RPP perovskite electronic devices with graphene as the electrical contact had a much better performance than those with Au as the electrodes. In addition to some intrinsic differences, including the different work functions and tunneling gap between graphene and Au, the extrinsic factors, such as the different surface roughness and thickness between graphene and Au, may also contribute to the variation in device performance. Typically, Au has a rougher surface than atomically flat graphene, which will give rise to a poorer contact quality. And usually the thickness of Au electrodes is around 50-100 nm. Considering the fragile mechanical properties of RPP halide

perovskite, it is very easy to generate some defects, sometimes even fractures, in RPP perovskites when transferring a few layers of RPP perovskites onto pre-patterned Au electrodes. Please elaborate the role from the extrinsic factors in the main text, and discuss which should be the dominating factor in determining the performance of the related RPP perovskite devices.

ANS: The referee is right to point out that besides intrinsic factors (eg. Work function, tunnelling gap differences), extrinsic factors may play a role. It is certainly true that on a rougher surface such as evaporated gold, the performance can be compromised.

We compared the tunnelling gap of RPP when it is supported on atomically smooth single crystal gold (Fig. R1a), versus graphene. Even on atomically smooth single crystal gold where the atomic structure is imaged by STM, we observed that the tunnelling gap is larger for RPP/Au as compared to RPP/graphene.

Fig. R1| **a**, Atomic structure of Au single crystal substrate used as substrate for monolayer and few-layer RPP. Scale bar; 1.5 nm. Comparison of tunneling gap of $n = 4$ RPP on single crystalline Au substrate (**b**) and graphene substrate (**c**) for STS measurements.

To reduce the contribution by extrinsic factors, we have endeavoured to use smooth gold electrodes with roughness about of 0.7 nm for device fabrication, as determined by AFM in the fabrication of RPP/Au devices (Fig. R2). We have also carefully ensured that the transferred perovskites are free from visible defects and cracks (see the damage-free optical image of transferred RPP on Au in Fig. R3 below). Therefore, we do not think roughness play a major role in the device performance, and it should be intrinsic charge injection barriers arising from interfacial energy alignments that play the most important role.

Fig. R2| **a**, AFM topography image of a Ti/Au film (4/45 nm) prepared by electron beam evaporation. **b**, Height profile along the white line in **a**.

Fig. R3 | Optical images of molecularly thin RPP flake on hollow Au electrode (50 nm thickness) by dry transfer in glove box.

Correction: we have added following description in Page 7 of revised manuscript.

“Besides the extrinsic factor of surface roughness on gold substrate, we believe the major factor is due to the higher injection barrier on gold compared to graphene. The van der Waals interface between G and RPP interface also benefits the performance. By comparing output characteristics at 45K to 100 K for both G and Au-contacted RPP devices (Fig. R4, below; Extended Data Fig. 9), the larger hysteresis present in the $I(V)$ curves of RPP/Au device may originate from the inhomogeneity of RPP on Au surface.”

Fig. R4 | Output curves of RPP-G FET at **a**, 45 K; **b**, 60 K and **c**, 100 K. Output curves of RPP-Au FET at **d**, 45 K; **e**, 60 K and **f**, 100 K.

3. It seems that the deviation between devices is significant. As was claimed by the authors, there should be a clear Schottky barrier between graphene and RPP. However, the barrier seems to disappear in Extended Data Fig.11a. In addition, please explain why there is an open-circuit voltage and short circuit current for the red curve in in Extended Data Fig.11b ($346 \mu\text{W}$, $V_g = -50 \text{ V}$).

ANS: For the curves in Extended Data Fig.11a, the Schottky barrier did not disappear. For clarity, we have plotted the curve in dark separately (Fig. R5). Apparently, the device exhibits non-linear current-voltage characteristics indicative of a Schottky barrier. Nevertheless, we do observe that the

current-voltage characteristics become increasingly linear with increasing laser power. This is due to photo-excited charge carrier generation, which largely reduces the apparent Schottky barrier height and the contact resistance. This observation is in agreement with the result in a recently published paper [Wang, Y. et al; Probing photoelectrical transport in lead halide perovskites with van der Waals contacts. *Nature Nanotechnology*. 2020].

Fig. R5 |a, I-V curves of the phototransistor in the dark and under different illumination intensities at $V_g = 50$ V. b, The corresponding I-V curve under dark in a.

Besides, we note that the observation of an open-circuit voltage and short circuit current in Extended Data Fig.11b is a result of the misleading double y-axis. In fact, for both the curves in dark and under illumination, they cross their respective (0, 0) (Fig. R6, below). We have re-plotted the data with arrows and aligned $Y=0$ for both curves for improved clarity shown in revised supporting information.

Fig. R6 | Re-plotted output curves of RPP phototransistor in the dark and under light illumination at $V_g = -50$ V (b) and $V_g = 0$ V (c).

4. As shown in Extended Fig 11c, the RPP/graphene devices is very insulating even under a very strong illumination of 346 μ W. This is much lower than the RPP/Au devices the authors' previous report in Nature Materials. Based on this comparison, the Au seems to have much better contact than graphene, which is opposite to the conclusion for this manuscript.

ANS: The referee may have looked at the wrong axis in Extended Fig 11c. Please look at the red axis (light-on current), not the blue one (dark current) in Fig. R7b below, which is Extended Figure 11c. Compared to Fig. R7a, which is the photodetector performance based on molecularly thin RPP on Au reported in Nature Material, the current for RPP/G device is clearly almost twice that for RPP/Au.

Fig. R7 | **a**, Photodetector performance based on molecularly thin RPP-Au device as shown in our previous published paper in Nature Materials. **b**, Photo-transistor performance of molecularly thin RPP-G device under $V_g = 0$ V (Extended Figure 11c).

We have revised the Extended Figure 11.

5. From the discussion in the main text and Fig. 1f, the $n=4$ RPP halide perovskite is highly n doped as the fermi level is very close to the conduction band minimum. However, this is counterintuitive as the intrinsic electrical conductivity of the $n=4$ RPP halide perovskites without gating and illumination is very low (from this manuscript and previous reports). If possible, please estimate the carrier concentration of $n=4$ RPP halide perovskite based on the electrical measurement results.

ANS: Based on the comments of the referee, we have re-examined how we arrived at the n -doping fermi level. This relies on estimating the position of the valence band edge with respect to the fermi level, and then accounting for the electronic band gap determined from ellipsometry. The corrected E_c-E_f is about 0.4 eV (Fig. R8a) which is quite deep for a donor level, and coupled with the reported low effective density of states (N_c and N_v) for perovskites, the carrier concentration is quite low at zero gate voltage condition. We have adjusted our energy level plot (Fig. R8b). We have corrected Fig. 1e and f in revised manuscript.

Fig. R8 | **a**, Valence band spectroscopy of $n = 4$ RPP showing valence band edge. **b**, Interfacial energy alignment diagrams of $n = 4$ RPP, graphene and Au.

In this work, the carrier density can be calculated from applied gate voltage V_g assuming a parallel-plate capacitor model¹, $n = (V_g - V_{th})\epsilon_0\epsilon_r/d$, where V_{th} is the threshold gate voltage, $\epsilon_0 = 8.85 \times 10^{-12}$ F/m is the vacuum permittivity, $d = 300$ nm is the thickness of the SiO_2 gate dielectric and $\epsilon_r = 3.9$ is its dielectric constant. In Fig. 3a (also shown as Fig. R9 below), V_{th} can be determined to be 34.7 V by extrapolating the linear region of the transfer characteristics to zero drain current² (forward scan curve is used here). Consequently, the carrier density is $n = 7.2 \times 10^{10} \times (V_g - 34.7)/\text{cm}^2$. Take $V_g = 60$ V for example, its value will be $1.82 \times 10^{12}/\text{cm}^2$.

Fig. R9 | Transfer characteristics of RPP-G FET at 1.7 K.

6. The reference citations are out of date. Several highly relevant works are not mentioned in the introduction. For example, regarding the graphene and 2D perovskite heterostructure, a recent electrical study (Joule 2018, 2, 2133) and a comprehensive review (Chemical Society Reviews 2018, 47, 6046-6072) should be cited. Regarding charge injection from Au electrode to 2D perovskites and related FET performance, a recent studying incorporating conjugated organic ligands (JACS 2019, 141, 15577-15585) should be mentioned.

ANS: We would like to thank the referee for highlighting this paper to us. “Ultrasensitive Heterojunctions of Graphene and 2D Perovskites Reveal Spontaneous Iodide Loss” (Joule 2018, 2, 2133). It should be pointed out that this paper is about graphene phototransistor whose performance is influenced by the chemical integrity of 2d perovskites interfaced to it, therefore it is fundamentally different from our paper that focused on FET fabricated on 2d perovskites. Nonetheless we have cited it as reference 8.

— Figure is cited from the paper of *Joule* 2018, 2, 2133

Correction: The review (Chemical Society Reviews 2018, 47, 6046-6072) and the paper (Joule 2018, 2, 2133) were cited as reference 7 and 8 with added description of “Heterostructure of 2D perovskite with other 2D materials is useful for fabricating functional devices^{7,8}.” in Page 2 of revised manuscript.

The paper (JACS 2019, 141, 15577-15585) has been cited as reference 14 with added description of “To date, FET mobilities of perovskites were mainly reported for spin-coated 3D perovskites¹¹⁻¹⁴ and Sn(II)-based 2D perovskites films¹⁵⁻¹⁷.” in Page 2 of revised manuscript.

7. Please double check the maximum laser power density, which was 346 uW in the figures but 348 uW in the main text.

ANS: Thanks. We have corrected 348 to 346 μW in the main text.

Response to reviewer 2

Reviewer #2 (Remarks to the Author)

The manuscript reports the study of the 2D RPP/graphene heterojunction through FET performance, STM/STS and femtosecond pump-probe characterization. It contains some interesting results, with the STM studies showing distinct morphology on different substrates. However, it is not clear at this point what changes the FET device performance as the graphene and Au contact are prepared using different techniques, note the top surface of Au after deposition can be rather rough. So based on the current data, it is hard to say whether the differences come from the van der Waals contact or just because of the change in metal work function. The control experiment present in the work contains two varying factors. For a control experiment, it would be more proper to use Ag whose work function is close to 4.5eV. On the other hand, although the graphene contact shows lower tunneling barrier compared with Au, which is $\sim 0.6\text{eV}$, but the value itself seems to have room for improvement.

ANS: We agree with the referee's comments and will respond point by point.

"However, it is not clear at this point what changes the FET device performance as the graphene and Au contact are prepared using different techniques, note the top surface of Au after deposition can be rather rough."

ANS: There are both extrinsic and intrinsic effects that affect the performance of FET devices. Besides intrinsic effects such as work function (which favours lower injection barrier for G over Au), we are well aware that extrinsic effects such as roughness may play a role.

To prove that the intrinsic factor has a more dominant role in controlling the tunnelling barrier, we have prepared RPP on single crystalline Au with the same method of preparing RPP/G. The single crystalline Au has been verified by STM to be atomically smooth (Fig. R1a). As shown by the scanning tunnelling spectroscopy data in Fig. R1b and c, the tunnelling barrier of RPP/Au is higher than that of RPP/G for the RPP with same thickness. The size of the RPP domains on G is also larger than Au, attesting to the fact that van der Waals interface supports a higher degree of structural order for RPP.

Fig. R1| **a**, Atomic structure of Au single crystal substrate used as substrate for monolayer and few-layer RPP. Scale bar; 1.5 nm. Comparison of tunneling gap of $n = 4$ RPP on single crystalline Au substrate (**b**) and graphene substrate (**c**) for STS measurements.

To reduce the contribution by extrinsic factors, we have endeavoured to use smooth gold electrodes with roughness about of 0.72 nm for device fabrication, as determined by AFM in the fabrication of RPP/Au devices (Fig. R2). We have also carefully ensured that the transferred perovskites are free from visible defects and cracks (see the damage-free optical image of transferred RPP on gold in Fig.

R3 below). Therefore, we do not think roughness play a major role in the device performance, and it should be intrinsic charge injection barriers arising from interfacial energy alignments that play the most important roles.

Fig. R2 | a, AFM topography image of a Ti/Au film (4/45 nm) prepared by electron beam evaporation. Scale bar; 400 nm. b, Height profile along the white line in a.

Fig. R3 | Optical images of molecularly thin RPP flake on hollow Au electrode (50 nm thickness) by dry transfer in glove box.

The control experiment present in the work contains two varying factors. For a control experiment, it would be more proper to use Ag whose work function is close to 4.5eV.

ANS: We do not think it is necessary to do this as Ag surface is more easily oxidized compared with Au to make situation more complex. Moreover, Ag surface is even rougher than Au, more interfering factors will contribute to poor device performance.

1) How is the graphene Fermi level determined to be 4.5eV?

ANS: We determined the work function of graphene using Kelvin Probe measurements. The graphene was placed on gold electrode during the measurement, where the work function of gold was pre-calibrated. It has been reported that with Pd or Au as contact, the work function of graphene assumes a value of ~ 4.62 eV by detailed analysis of the capacitance voltage (C-V) of a metal-graphene-oxide-semiconductor capacitor structure [Song, S.M. et al; Determination of work function of graphene under a metal electrode and its role in contact resistance, *Nano Lett.* 2012, 12, 8, 3887-3892]. Our determined value of ~ 4.5 eV has been calibrated with standard gold samples (Fig. R4, below) and is close to reported values in literature^{3,4}. We are aware that there can be a difference of ~ 60 meV between monolayer graphene and multilayer graphene, on account of charge transfer between graphene and substrate where charge distribution is found to decay exponentially from substrate, thus there is an error margin of ~ 60 meV when we report a work function of 4.5 eV

[Leenaerts, O. et al; The work function of few-layer graphene, J Phys Condens Matter. 2017;29(3):035003].

Fig. R4 | KPFM measurements of G on Au substrate.

Correction: we have cited the mentioned papers in revised SI.

2) It is strange that the Fermi level doesn't change with layer number at all but the band gap varies with layer number as shown in the STS measurements. Based on the band analyses in Fig. 1, the monolayer RPP should have fairly high electron density so that narrow tunneling barrier width, what is the basic two-terminal IV looks like under dark.

ANS: First we have to clarify that what we measured in Scanning tunnelling spectroscopy (STS) is the tunnelling gap and not the real electronic gap. Electron tunnelling from the valence bands may leave behind holes, if these holes did not travel to the bottom electrode fast enough or get trapped (especially for thick layers), a space charge layer that is thickness- dependent may form, leading to the shift in the onset voltage. The apparent energy gap measured here is a convolution of the single electron band gap and tunnelling barrier between the substrate and RPP.

We did not do comprehensive experiments to establish whether the fermi level changes with layer number, although we have measured the monolayer, bilayer and bulk to found the work function to be similar. One reason is that work function measurement is sensitive to surface electrostatic charges. 2D perovskite is multiple quantum well structure segregated by layers of dielectrics (organic chains), so work function measurement may be essentially measuring the top most layer, independent of the layer thickness. The band gaps did not change significantly from a band structure point since the hybridization between the organic layers are prevented by the insulating chains. However, the tunnelling gap, as opposed to the electronic band gap can be affected by the thickness as explained by the reasons above.

Monolayer RPP is fairly insulating in the dark. The Fig. R5 below shows the basic two-terminal IV curve under dark. Though monolayer RPP is n-doped, the donor level is quite deep ($E_c - E_f = 0.4$ eV) (Fig. R6), we have made a mistake in drawing the fermi level to close to the conduction band edge and corrected it in revised manuscript.

Fig. R5 | **a**, I(V) curve of RPP-Au FET at $V_g = 0$, 45 K. **b**, I(V) curve of RPP-G FET at $V_g = 0$, 45 K.

Fig. R6 | **a**, Valence band spectroscopy of $n = 4$ RPP showing valence band edge. **b**, Interfacial energy alignment diagrams of $n = 4$ RPP, graphene and Au.

3) Considering the large hysteresis in the FET transfer curve, how is the mobility determined under such circumstances. And what is the more fundamental reason for FET mobility increase with decreasing temperature.

ANS: Given the large hysteresis, we should extract mobility on both forward and reverse sweepings. In the revised manuscript, we have added the mobility versus temperature (Fig. R7, below) derived from forward sweeping shown in Extended data Fig. 10b in revised SI.

Fig. R7 | Temperature dependent field-effect electron mobility extracted from the forward and reverse transfer characteristics.

At higher temperature, the dynamic rotation of MA^+ ions and increased electron-phonon coupling reduces the mobility^{5,6}. Fröhlich interactions are the dominant intrinsic mechanism limiting charge-carrier mobility in polar crystals such as perovskites. For example, electron-phonon coupling was found to be markedly reduced below the characteristic temperature corresponding to the energy of the relevant LO phonon (11 meV), in agreement with Fröhlich interactions being the dominant mechanisms at room temperature. In $MAPbI_3$ perovskites for example, there is a $T^{-1.5}$ temperature dependence of the charge-carrier mobility in the high-temperature regime⁷. Hence a similar behaviour is expected here for the quasi-2D perovskites.

4) The authors mentioned “The value of V_{trans} is related to the barrier height for charge injection”, but as the slope of Fig. 3c doesn’t change between each curve, so the barrier height seems to be constant, the authors need to explain this part.

ANS: We admit that at first glance, the slopes of the three curves look very similar in Fig. 3c which is due to the scale range of the plot. However, there are significant differences in these three slopes.

Table R1 below shows the slope result extrapolated by linear fitting of Fowler-Nordheim tunnelling regime at different V_g . It is noted that the slope (absolute value) has decreased as V_g increasing, which is consistent with the trend of V_{trans} .

V_g (V)	Slope
20	-6.39 ± 0.26
40	-5.36 ± 0.03
65	-3.98 ± 0.03

Table R1 | The slope results by linear fitting of Fowler-Nordheim tunnelling regime at different V_g shown in Fig. 3c.

Correction: We have added this table as Extended Table 2 with related description “The slope value by linear fitting of F-N tunnelling regime decreases as V_g increases (Extended Table 2) which agrees with the trend of V_{trans} .” in revised manuscript of Page 9.

5) Comparing Fig. 3b and 3f, the 75K dark transfer data showing at least an order difference, please explain the discrepancy. Why does sample under illumination show weak field effect, especially for weak illumination case where the current is not large (i.e. the carrier density is not high). Additionally, the illumination intensity is suggested to be normalized in the format of power/area, which would allow the readers to compare with others’ work easier.

ANS: These differences in dark current are due to different sample treatment procedures. For FET devices, we have added a pre-annealing step before measuring. As shown in Fig. R8 below, after a few minutes of pre-annealing, I_{ds} has increased by one order. To be consistent, a pre-annealing step of 60 degree for 5 mins was conducted on our FET device in high vacuum before measuring.

Fig. R8 | A pre-annealing step of RPP FET in high vacuum before measuring.

For photo-FET devices, we did not do pre-annealing step due to several reasons. 1, we want to explore pure laser effect without any other interference. 2, To be consistent with our previous reported RPP photodetector (Nat. Mater. 17, 908-914 (2018)). The reported I_{dark} ($V_g = 0$, Extended Data Fig. 11) in this paper is around 10^{-12} to 10^{-13} which is quite consistent with our previous reported data. In fact, for photodetector/photo-FET we expect low dark current. 3, As shown in Fig. 3e, under $346 \mu W$, $V_g = 50 V$, $77K$, the photo-FET performance ($\sim 1100 nA$) is even a higher than FET in Fig. 3b under same condition ($\sim 300 nA$), thus pre-heating step can be ignored in photo-FET.

Correction: we have added some description in revised manuscript of Page 7 to make it clear. “The FET devices were heated at 60 degrees for 5 min in high vacuum before measurement.”

The field effect is still present although it has been reduced when laser shines on the sample, this can be understood because more carriers are generated that screen the field. When we plot out the

data and magnifies the scale (Fig. R9), we can see the field effect clearly for irradiance between 28-1100 W/cm², as shown below:

Fig. R9 | Transfer characteristics under different illumination intensities for RPP/graphene device at 77 K.

The photo-FET devices were measured under focused laser with 532 nm excitation (1 μm² spot area). In this case, an illumination condition of power of 1.7 μW is equivalent to a power/area of 170 W/cm². We have normalized the illumination intensities into the format of power/area indicated in Fig. R9 above. Compared with published papers, the illumination intensities here are not very weak and this may be the reason why field effect is not very obvious. We also added the description of “Upon photo-irradiation with energy larger than the quasi-particle gap (focused laser with 532 nm excitation; 1 μm² spot area)” in page 9 of revised manuscript.

6) Extended Fig.6 suggests G/Au based devices show different asymmetries in the transfer characteristics. Does it mean the carrier type is different, please explain.

ANS: Yes. for RPP on G electrode, the FET exhibits dominant *n*-type behaviour. By contrast, RPP-Au FET shows dominant *p*-type behaviour. This has its origin in the relative Fermi level (E_F) positions and the direction of interfacial charge transfer. As shown in Fig. 1f of manuscript, E_F of G ($E_{F,G}$) is in a higher position than E_F of RPP ($E_{F,RPP}$) allowing electrons to be injected from G to RPP. This phenomenon will be more prominent when V_g has been applied. G is *n*-doped⁸⁻¹⁰ at positive V_g indicating the $E_{F,G}$ position is even higher than $E_{F,RPP}$. The enhanced E_F difference causes more electrons to be injected into the RPP from G leading to high I_{ds} . As observed in Fig. 3a and 3b, I_{ds} increase a lot as V_g increase in positive direction (from 20 V to 65 V).

The opposite situation applies in the case of RPP-Au FET. As observed in Fig. 3a and Extended Data Fig. 6 d-f, I_{ds} increase as V_g increase in negative direction (from -20 V to -65 V) which is attributed to the E_F of Au ($E_{F,Au}$) is in a lower position compared with $E_{F,RPP}$ leading to RPP *p*-doped.

Correction: We also added the description of “Moreover, electrode dependent transport behavior is related to Fermi level alignment of RPP, G and Au initially shown in Fig. 1f. As V_g increasing in positive direction, G is *n*-doped and enlarged Fermi energy difference between G and RPP facilitate electrons transfer from G to RPP. In contrast, electrons transfer from RPP to Au can only happen in Au-contacted device.” in page 8 of revised manuscript.

7) Based on the current analyses, it is hard to evaluate the performance of the 2D RPP/graphene FET of this work with other lead halide perovskite FETs reported in the literature, a table including on/off ratio, on state current, geometry, FET mobility.... is suggested to be added.

ANS: Thanks for the good suggestion. We have summarized an Extended Table 1 of reported hybrid perovskite FETs including both 2D and 3D. Our mobility of lead based 2D perovskite is close to a few reported polycrystalline 3D $\text{CH}_3\text{NH}_3\text{PbI}_3$ FETs.

Correction: We have included this table in SI.

Extended Table 1 | A summary of reported FET works based on (hybrid) lead halide perovskite semiconductors.

Material	Preparation method	L/W (μm)	S/D contact	Mobility (cm^2/Vs)	Regime for Mobility extraction	On/off ratio	Ref
Bilayer (7 nm) $(\text{C}_4\text{H}_9\text{NH}_3)_2(\text{CH}_3\text{NH}_3)_3\text{Pb}_4\text{I}_{13}$	Exfoliation	L=0.3 W=8.2	G	0.1 (e) @1.7 K	Linear	1.6×10^6	This work
30 nm polycrystalline $(\text{C}_6\text{H}_5\text{C}_2\text{H}_4\text{NH}_3)_2\text{SnI}_4$	Spin coating	L=28 W=1000	Pd, Pt, Au	0.6 (h) @RT	Saturation	10^4	11
Polycrystalline $(\text{PEA})_2\text{SnI}_4$	Melt-process	L=105 W=1000	Au	1.7 (h) @RT	Saturation; Linear	10^6	12
150 nm Polycrystalline $\text{CH}_3\text{NH}_3\text{PbI}_3$	Spin coating	L=80/100 W=20000	Ni/Au	0.067/0.072 (e); 0.0066/0.021 (h) @78K	Linear/ saturation	10^5 @198 K	5
100 nm $(\text{CH}_3\text{NH}_3\text{PbI}_3)$; mixed-halide $(\text{CH}_3\text{NH}_3\text{PbI}_{3-x}\text{Cl}_x)$	Spin coating	L=50 W=1000	Ti/Au	0.18(h)/0.17(e); 1.24(h)/1.0(e) @RT	Saturation	10^4	13
$\text{CH}_3\text{NH}_3\text{PbI}_{3-x}\text{Cl}_x$	Spin coating	L=30 W=1000	Au	1.3(h); 1(e) @RT	Saturation	10^2	14
40 nm polycrystalline PEASnI_4	Spin-coating	L=95/45 W=2000	Au	0.53 to 15 (h) @RT	Saturation	10^6	15
Polycrystalline $\text{CH}_3\text{NH}_3\text{PbI}_3$; Single crystal $\text{CH}_3\text{NH}_3\text{PbBr}_3$	Vapor grown; anti-vapor diffusion		Au	8 (μ_{hall}) 60 (μ_{hall}) @RT	Hall bar		16
1 mm $\text{CH}_3\text{NH}_3\text{PbBr}_3$ single crystal	Anti-vapor diffusion	L=750 W=1300	Ti	10 (μ_{hall}) @ 300 K	Hall bar		17
40 nm Polycrystalline $(\text{C}_6\text{H}_5\text{C}_2\text{H}_4\text{NH}_3)_2\text{SnI}_4$	Spin-coating	L=95 W=2000	Au, Ag, Al	0.6/1.1/1.5 (e) @RT	Saturation	5.2×10^4	18
100 nm polycrystalline $\text{CH}_3\text{NH}_3\text{PbI}_3$	Spin-coating	L=10 W=1000	Au	0.02 (e) @RT; 0.6 (e) @100 K; 0.05 (e) @270 K	Saturation	10^5	6
250 nm $\text{Cs}_x(\text{MA}_{0.17}\text{FA}_{0.83})_{1-x}\text{Pb}(\text{Br}_{0.17}\text{I}_{0.83})_3$	Spin-coating	L=20 W=1000	Au	2.1 (h) @RT 2.5 (e) @RT	Saturation	10^4	19
MAPbX_3 (X=Cl, Br, I)	TSCs fabrication	L=265/240/ 185 W=50	Au	1.8 (h); 1.9 (h); 1.5 (h) @RT	Saturation	10^4 ; 10^3 ; 10^3	20
400 nm polycrystalline MAPbI_3	Hot-casting	L=70 W=100	Au	0.001 (h) @RT	Saturation	10^4	21
RbCsFAMAPbI_3	Spin coating	L = 100 W = 1000	Cr/Au	1.2 (e) @RT	Saturation	10^4	22
$(\text{PEA})_2\text{SnI}_4$	Spin coating	L=200 W=1000	Au	3.51 (h)	Saturation	3.4×10^6	23
$(4\text{Tm})_2\text{SnI}_4$	Spin coating	L=40 W=2880	Au	2.32 (h,max)	Linear	10^5 - 10^6	24

8) Please mark the domain sizes in Fig. 2 based on the discussion “on graphene or Au (Fig. 2c and 2d) can have a single domain with lateral width >200 nm, but the size of the domain decreases to 100 nm and 20 nm in bilayer and monolayer (Fig 2a and 2b)”.

ANS: As shown in Fig. R10 below, the single crystal domain size is increased as the thickness of RPP has increased. This trend has been confirmed by repeating STM measurements. In Fig. R10a, it exhibits domain size width of 10-20 nm in monolayer. For bilayer, the width of domain size has increased to about 100 nm (Fig. R10b,c). We have to make a correction here: for bulk RPP, as the size of the domain is outside of the scanning range of STM, that is why we indicate as > 200 nm, we did not manage to see the grain boundaries. It is a bit subjective statement of bulk here. In revised manuscript, we have corrected to highlight the trend.

Correction: In page 5, "STM studies reveal that the lateral width of the single domain increases with the thickness of RPP supported on G or Au (Fig. 2a-e and Extended Fig. 5a,b,d). For example, the domain width increases from ~ 15 nm in monolayer (Extended Data Fig. 5a) to ~ 100 nm in bilayer on G (Extended Data Fig. 5c)."

Fig. R10 | The domain size of monolayer (a), bilayer (b,c) RPP and bulk (d) on graphene. The domain boundaries are outlined by the dash line and the orientation of the dimer alignment is indicated as black arrow in the STM images.

Response to reviewer 3

Reviewer #3 (Remarks to the Author)

In this manuscript, Leng and coauthors introduce graphene to contact molecularly thin 2D perovskite plane so that the contact barrier is reduced and the mobility of 2D perovskite is improved. Due to the presence of long organic chains in 2D perovskites, the electrical conductivity in the vertical direction is rather poor, leading to the poor performance for 2D perovskite based electronic devices.

The lowered contact barrier in 2D perovskite/graphene interface thus can partially solve the charge injection issue in 2D perovskite electronic devices. In particular, STM study reveals that highly ordered domain structure can be sustained down to a monolayer for 2D perovskite plate on graphene while disorders can be observed for monolayer 2D perovskite plate on Au. This study is interesting and important for 2D perovskite community. However, the following my concerns are needed to addressed before I can recommend its publication.

1 The authors claimed that from STM study, a thick 2D perovskite crystal supported on graphene or Au can have a single domain with lateral width >200 nm, but the size of the domain decreases to 100 nm and 20 nm in bilayer and monolayer. Is it possible that even in thick sample the size of the domain is still around 20 nm like monolayer case since the interlayer coupling in 2D perovskite is rather weak? Also, can the authors comment how the domain size affects the transport properties of the graphene contacted 2D perovskite devices?

ANS: We have repeated the experiments again on Au (Fig. R1, below) and G (see Fig. R10 above for reviewer 2), and indeed, the results confirmed that thicker perovskites have larger single crystal domains compared to the thinner ones. The explanation is this: Although the bulk is centrosymmetric and non-polar, at the surface, polar discontinuity exists and the surface atoms may show some polarization. As the crystal gets thinner, the screening field from the substrate (metallic) will have a depolarization effect and that is why the domains appear smaller and more random.

Usually, when the domain size has decreased, the field effect mobility decreased⁶. This is also applied to our molecularly thin 2D perovskites. As a temperature-induced structural phase transition has been suppressed in molecularly thin level²⁵, domain size plays an important role for FET performance with same substrate. This may be the reason why our monolayer RPP (n = 4) FET shows lower performance compared with bilayer one.

Fig. R1 | STM images of thick layer RPP on Au substrate, showing the larger domain size compared with thin RPP (Fig. 2e, manuscript).

2 In Figure 1d, the authors extracted the bandgap of 2D perovskite by extrapolated the edge of absorption spectrum. However, due to the strong excitonic effect, this method might be not applicable here since the exciton states with large n would also contribute to the absorption edge.

ANS: although there is a strong excitonic effect at the edge of absorption spectrum, the difference between excitonic peak and band gap edge can still be well distinguished. In fact, in the manuscript, the band gap energy is obtained by extrapolating from the band gap edge. The excitonic features are actually located at lower energy, which will not affect the extrapolation of band gap.

To further clarify this question, we also fit the spectrum as shown in Fig. R2. We use Lorentz functions to fit the excitonic peaks and Tanguy function for the band edge. The fitting results show the band gap of sample is 2.07 eV, which is similar as we extrapolate from band gap edge.

Fig. R2 | Experimental and fitting results of imaginary part of dielectric function.

3 Can the authors comment how the degradation of 2D perovskite affects their results? Previous study has revealed that despite the stability of 2D perovskite has been improved, they still undergo quick degradation. Under such case, it is evitable that degradation would take place during the dry transfer process. With the degradation of 2D perovskite, surface depletion field might be present (see Nano Research 12, 2858–2865(2019)). The surface depletion field possibly effect the STM image.

ANS: Our STM is conducted in ultrahigh vacuum. The samples are exfoliated and dry transferred in glove box., Moreover, we have vacuum transferred through a portable vacuum transfer station. If the samples show any sign of degradation, STM will see very rough and random features. In contrast, when the surface is near pristine, we will observe uniform and smooth domains with atomic resolution. Thus, we do not think the samples have degraded in anyway.

4 The authors attribute the hysteresis in transfer curves to the trap-dominated electronic mechanism while no field effect above 250 K to the screening of ion migration. Under such case, how did the authors exclude the influence of ion migration on the hysteresis.

ANS: At the temperature range where we report the FET mobility, spanning 1.7K-75K (regime I), the IV curves do not show significant hysteresis (Fig. R3a-c). Therefore, there is minimal ion migration.

At Regime II, which is between 100K-175K, the increased hysteresis in both IV curves (Fig. R3d-f) and IV_g curves (Extended Data Fig. 10) are attributed to trap state and surface dipoles^{5,26}.

At Regime III, which is between 225K-300K, ion migration causes strong hysteresis⁶.

Fig. R3 | Output curves of RPP-G FET at **a**, 45 K; **b**, 60 K; **c**, 75 K; **d**, 125 K; **e**, 150 K; **f**, 175 K; **g**, 225 K; **h**, 250 K and **i**, 300 K.

Correction: We have added the figure as Extended Fig. 9 in revised manuscript.

5 Since a large hysteresis has been observed, the authors are needed to specific how did they extract mobility. A different mobility will be obtained from backward and forward scanning. In addition, the authors are suggested to comment how the hysteresis would affect the precise evaluation of mobility.

ANS: Thanks for the comment. We agree that more information on mobility extraction should be given. In the manuscript, the mobility was derived based on the backward sweep of V_g (from positive to negative).

Indeed, as mentioned by the reviewer, the large hysteresis should give rise to different mobilities on both forward and reverse sweeps. Therefore, we have calculated both in the revised manuscript. Despite the slight difference between them, they do show similar trend with temperature.

The hysteresis in transfer characteristics may result in systematic errors in mobility extraction. Typically, for the observed clockwise hysteresis, the backward sweep tends to overestimate the electron mobility whereas the forward sweep might underestimate its value [Li, D., et al.; Size-dependent phase transition in methylammonium lead iodide perovskite microplate crystals. Nature Communication 7, 11330 (2016)].

In the revised SI, we have added the mobility versus temperature (Fig. R4, below) derived from forward sweeping shown in Extended data Fig. 10b.

Fig. R4 | Temperature dependent field-effect electron mobility extracted from the forward and reverse transfer characteristics.

6 Apparently, the contact is non-Ohmic either in dark or under light illumination.

ANS: We agree that we should not use the term “Ohmic contact”. We have revised the description to “Transport studies show linear *I*-*V* characteristics” instead of “Ohmic contact”.

The current-voltage characteristics become increasingly linear with increasing laser power. This is due to significant charge carrier generation, which largely reduces the apparent Schottky barrier height and the contact resistance. This observation is in agreement with the result in one recently published paper [Wang, Y. et al; Probing photoelectrical transport in lead halide perovskites with van der Waals contacts. *Nat. Nanotechnol.* 2020]. This paper has been cited as reference 37 in revised manuscript.

Correction: In page 9 of revised manuscript “the output curve of RPP/G photo-FET device shows a non-linear (blue)-to-linear *I*-*V* behaviour (red) transition, as shown in Fig. 3e and Extended Data Fig. 14a. Photo-irradiation creates photocarriers and reduces the Schottky barrier width, thus reducing interfacial resistance³⁷.”

7 How the light illumination affects the hysteresis in transfer curves?

ANS: In this work, the hysteresis reduces with light illumination (Fig. R5, below). This can be ascribed to the dominating conduction of large amount of photogenerated carriers generated under high illumination intensities which reduces channel resistance drastically.

Fig. R5 | Transfer characteristics in the dark and under different illumination intensities for RPP-G device at 77 K.

8 The authors attribute the increase of current under light illumination to the exciton dissociation. Nevertheless, the exciton binding energy is estimated to be over 100 meV from Figure 1d. Under such case, the exciton dissociation cannot occur at room temperature.

ANS: The referee is right to point out that the exciton binding energy for 2D perovskites is too high to allow thermal dissociation at room temperature. However, such phenomena have been observed for TMD materials such as WSe_2 , where despite the large exciton binding energy, a high photoresponse rate has been observed [Massiotte, M. et al, Dissociation of two-dimensional excitons in monolayer WSe_2 , *Nature Communications* (2018) 9: 1633]. Theoreticians have explained this using a model where electric fields provide the energy needed to dissociate the excitons, especially for ultrathin crystals where a high in-plane field is experienced [Sten Haastrup, et al, Stark shift and electric-field-induced dissociation of excitons in monolayer MoS_2 and h BN/ MoS_2 heterostructures, *Phys. Rev. B* **94**, 041401(R) (2016); Benedikt Scharf, et. al, Excitonic Stark effect in MoS_2 monolayers, *Phys. Rev. B* **94**, 245434 (2016)]. Under static in-plane electric field, excitons dissociate at a rate corresponding to the tunnel ionization of 2D Wannier–Mott excitons [*Nature Communications* (2018) 9: 1633]. We can see that there is a rapid increase in photocurrent (photoresponse rate) as in-plane field is increased gradually at first, before it becomes more linear at higher in-plane field when the photoresponse rate becomes limited by drift-diffusive transport of the free carriers. Thus, at the low-field regime, the exciton dissociation process is the rate-limiting step governing the generation of photocurrent.

9 Have the authors measured the device performance of graphene contacted monolayer 2D perovskite?

ANS: The device performance of monolayer RPP is poorer compared with the few-layer one, we attribute this to the larger distortion exerted by the substrate on the monolayer RPP. We can see from STM studies that the domain sizes are smaller and the domain orientations are more random.

10 Can the authors comment what will change if we use different type of long organic chain in 2D perovskites?

ANS: If the length of alkylammonium cation is increased, the optical property of bulk crystals will not change much due to the weak interaction between layers. For example, in single crystal of $(\text{C}_m\text{H}_{2m+1}\text{NH}_3)_2\text{PbI}_4$ with $m = 4, 8, 9, 10$ and 12 , they show a similar 320 ± 20 meV exciton binding energy²⁷. However, if changing aliphatic to aromatic amines, the optical property will change a lot due to different dielectric constant of organic cations. For example, there is a decrease in exciton binding energy from 320 meV in $(\text{C}_{10}\text{H}_{21}\text{NH}_3)_2\text{PbI}_4$ to 220 meV in PEA_2PbI_4 , consistent with PEA^+ having a higher dielectric constant than alkyl amines²⁸. The change of local environment of the perovskite also affect their optical property a lot. Upon exfoliation and transfer such that perovskite layer is bounded by substrate on one side and air on the other, the dielectric confinement created by the sample environment will affect band gap²⁵. All of these will leave us room to continue explore their electron and charge transport based on molecularly thin 2D perovskite.

References

- 1 Li, L. et al. Black phosphorus field-effect transistors. *Nature Nanotechnology* **9**, 372-377 (2014).

- 2 Ghibaudo, G. New method for the extraction of MOSFET parameters. *Electronics Letters* **24**,
543-545 (1988).
- 3 Panchal, V., Pearce, R., Yakimova, R., Tzalenchuk, A. & Kazakova, O. Standardization of
surface potential measurements of graphene domains. *Scientific Reports* **3**, 2597 (2013).
- 4 Liang, S.-J. & Ang, L. K. Electron Thermionic Emission from Graphene and a Thermionic
Energy Converter. *Physical Review Applied* **3**, 014002 (2015).
- 5 Chin, X. Y., Cortecchia, D., Yin, J., Bruno, A. & Soci, C. Lead iodide perovskite light-emitting
field-effect transistor. *Nature Communications* **6**, 7383 (2015).
- 6 Senanayak, S. P. *et al.* Understanding charge transport in lead iodide perovskite thin-film
field-effect transistors. *Science Advances* **3**, e1601935 (2017).
- 7 Herz, L. M. Charge-Carrier Mobilities in Metal Halide Perovskites: Fundamental Mechanisms
and Limits. *ACS Energy Letters* **2**, 1539-1548 (2017).
- 8 Liu, Y. *et al.* Toward Barrier Free Contact to Molybdenum Disulfide Using Graphene
Electrodes. *Nano Letters* **15**, 3030-3034 (2015).
- 9 Novoselov, K. S. *et al.* Electric Field Effect in Atomically Thin Carbon Films. *Science* **306**, 666
(2004).
- 10 Reddy, D., Register, L. F., Carpenter, G. D. & Banerjee, S. K. Graphene field-effect transistors.
Journal of Physics D: Applied Physics **45**, 019501 (2011).
- 11 Kagan, C. R., Mitzi, D. B. & Dimitrakopoulos, C. D. Organic-Inorganic Hybrid Materials as
Semiconducting Channels in Thin-Film Field-Effect Transistors. *Science* **286**, 945 (1999).
- 12 Mitzi, D. B. *et al.* Hybrid Field-Effect Transistor Based on a Low-Temperature Melt-Processed
Channel Layer. *Advanced Materials* **14**, 1772-1776 (2002).
- 13 Li, F. *et al.* Ambipolar solution-processed hybrid perovskite phototransistors. *Nature*
Communications **6**, 8238 (2015).
- 14 Mei, Y., Zhang, C., Vardeny, Z. V. & Jurchescu, O. D. Electrostatic gating of hybrid halide
perovskite field-effect transistors: balanced ambipolar transport at room-temperature. *MRS*
Communications **5**, 297-301 (2015).
- 15 Matsushima, T. *et al.* Solution-Processed Organic-Inorganic Perovskite Field-Effect
Transistors with High Hole Mobilities. *Advanced Materials* **28**, 10275-10281 (2016).
- 16 Chen, Y. *et al.* Extended carrier lifetimes and diffusion in hybrid perovskites revealed by Hall
effect and photoconductivity measurements. *Nature Communications* **7**, 12253 (2016).
- 17 Yi, H. T., Wu, X., Zhu, X. & Podzorov, V. Intrinsic Charge Transport across Phase Transitions in
Hybrid Organo-Inorganic Perovskites. *Advanced Materials* **28**, 6509-6514 (2016).
- 18 Matsushima, T. *et al.* N-channel field-effect transistors with an organic-inorganic layered
perovskite semiconductor. *Applied Physics Letters* **109**, 253301 (2016).
- 19 Yusoff, A. R. b. M. *et al.* Ambipolar Triple Cation Perovskite Field Effect Transistors and
Inverters. *Advanced Materials* **29**, 1602940 (2017).
- 20 Yu, W. *et al.* Single crystal hybrid perovskite field-effect transistors. *Nature Communications*
9, 5354 (2018).
- 21 Canicoba, N. D. *et al.* Halide Perovskite High-k Field Effect Transistors with Dynamically
Reconfigurable Ambipolarity. *ACS Materials Letters* **1**, 633-640 (2019).
- 22 Senanayak, S. P. *et al.* A general approach for hysteresis-free, operationally stable metal
halide perovskite field-effect transistors. *Sci. Adv.* **6**, eaaz4948 (2020).
- 23 Zhu, H. *et al.* High-Performance and Reliable Lead-Free Layered-Perovskite Transistors.
Advanced Materials **32**, 2002717 (2020).
- 24 Gao, Y. *et al.* Highly Stable Lead-Free Perovskite Field-Effect Transistors Incorporating Linear
 π -Conjugated Organic Ligands. *Journal of the American Chemical Society* **141**, 15577-15585
(2019).
- 25 Yaffe, O. *et al.* Excitons in ultrathin organic-inorganic perovskite crystals. *Physical Review B*
92, 045414 (2015).

- 26 Zaumseil, J. & Sirringhaus, H. Electron and Ambipolar Transport in Organic Field-Effect Transistors. *Chemical Reviews* **107**, 1296-1323 (2007).
- 27 Ishihara, T., Takahashi, J. & Goto, T. Optical properties due to electronic transitions in two-dimensional semiconductors $(C_nH_{2n+1}NH_3)_2PbI_4$. *Physical Review B* **42**, 11099-11107 (1990).
- 28 Hong, X., Ishihara, T. & Nurmikko, A. V. Dielectric confinement effect on excitons in PbI_4 -based layered semiconductors. *Physical Review B* **45**, 6961-6964 (1992).

REVIEWERS' COMMENTS

Reviewer #1 (Remarks to the Author):

The authors have revised the paper significantly and it is publishable now, in my opinion.

Reviewer #3 (Remarks to the Author):

I have no further questions and thus I would like to recommend its publication at the current form.